# The Uncertainty of COVID-19 Inducing Social Fear and Pressure on the Continuity of Rural, Community-Based Medical Education: A Thematic Analysis

**DOI:** 10.3390/healthcare9020223

**Published:** 2021-02-17

**Authors:** Ryuichi Ohta, Yoshinori Ryu, Chiaki Sano

**Affiliations:** 1Community Care, Unnan City Hospital, Unnan 699-1221, Japan; yoshiyoshiryuryu.hpydys@gmail.com; 2Department of Community Medicine Management, Faculty of Medicine, Shimane University, Izumo 693-8501, Japan; sanochi@med.shimane-u.ac.jp

**Keywords:** community-based medical education, COVID-19, family medicine, Japan, rural

## Abstract

Rural community-based medical education (CBME) enriches undergraduate and postgraduate students’ learning but has been impacted by the coronavirus disease 2019 (COVID-19) pandemic. We identified the challenges faced by stakeholders as well as the relevant solutions to provide recommendations for sustainable CBME in community hospitals during the COVID-19 pandemic. A total of 31 pages of field and reflection notes were collated through direct observation and used for analysis. Five physicians, eight nurses, one clerk, fourteen medical trainees, and three rural citizens were interviewed between 1 April and 30 September 2020. The interviews were recorded and their contents were transcribed verbatim and analyzed using thematic analysis. Three themes emerged: uncertainty surrounding COVID-19, an overwhelming sense of social fear and pressure within and outside communities, and motivation and determination to continue providing CBME. Rural CBME was impacted by not only the fear of infection but also social fear and pressure within and outside communities. Constant assessment of the risks associated with the pandemic and the implications for CMBE is essential to ensure the sustainability of CBME in rural settings, not only for medical educators and students but also stakeholders who administrate rural CBME.

## 1. Introduction

The coronavirus disease 2019 (COVID-19) pandemic has had a tremendous impact on community-based medical education (CBME). CBME offers undergraduate and postgraduate students multiple opportunities to learn and broaden their perspectives on medicine, practical knowledge and skills, and attitudes toward doctors, especially in rural settings [1,2,3]. In Asian contexts, the percentage of medical students who are positively motivated to work in rural areas varies from 32.8% to 69.2% [4], but less than a quarter of physicians actually work in rural areas [5]. CBME programs can motivate students to become family physicians and work in remote areas, which is critical for the sustainability of community and rural medicine [6,7]. In CBME, collaboration between various stakeholders such as universities, general hospitals, medical teachers, medical trainees, other healthcare professionals, and citizens is essential [1,8,9]. The COVID-19 pandemic, however, has the potential to deeply impact these relationships, primarily because of a mutual fear of infection.

Medical institutions fear the transmission of infection and the risk their trainees face in such a situation [10,11]. As a result of these institutions’ reduced involvement in clinical medicine, medical trainees may miss out on essential opportunities [10,12]. The continuity of CBME can be vital for the sustainability of rural medicine, as this education is what motivates medical trainees to work in rural institutions [13,14]. The COVID-19 pandemic has reduced the efficacy of CBME in Japan. Fear generated by the pandemic has led to a significant reduction in the availability of practical learning experiences that are essential for trainee doctors’ development, such as face-to-face interactions with patients and community workers [15]. Although online learning using social media and video conferences can be effective, some aspects of the curriculum can only be taught in person at medical institutions, which may not be taking place [16,17]. The severity of the pandemic varies depending on population density and the distance of cities from local points of origin [18,19]. In rural areas, there are few patients with COVID-19 and the standard precautions and protocols have been effective in controlling the infection [12,18,20].

The present adverse situation must be conquered to drive rural CBME and equip future medical trainees in rural community hospitals with adequate education [21]. In Japan, 97.5% of medical universities included CBME in their curricula; this has proved significant for trainees [22]. Thus, CBME is vital for Japanese medical education and should be continued. To continue efficiently delivering rural CBME, the perceptions of stakeholders need to be taken into consideration [12]. Therefore, in this study, the research question is as follows: What are the challenges that stakeholders face with regard to continuing to offer CBME, and how can these be resolved? Since this pandemic could continue indefinitely, such clarifications from stakeholders are critical for the sustainability of rural medicine [14]. This study fills a gap in the literature, as to date, there have been no studies of stakeholders’ perceptions of the continuity of rural CBME during the COVID-19 pandemic.

## 2. Materials and Methods

### 2.1. Setting

Unnan is a remote rural city in Japan. In March 2020, the total population was 37,637 (18,145 males and 19,492 females), of whom 39% were aged over 65. The city has 16 clinics, 12 homecare stations, 3 visiting nurse stations, and 1 public hospital. At the time the study was conducted, Unnan City Hospital had 281 care beds and 27 physicians, 197 nurses, 7 pharmacists, 15 clinical technicians, 37 therapists, 4 nutritionists, and 34 clerks. Most hospital health workers were from the city.

### 2.2. Rural CBME in Unnan City Hospital

In Unnan City Hospital’s CBME program, medical trainees work with family doctors at the community hospital and the affiliated clinic to learn about the most frequently reported illnesses and how they are managed through systematic practice and person-centered care—a comprehensive and integrative approach. They collaborate with care managers and home care workers to learn about interprofessional work, a competency expected of family doctors. In community care settings, they participate in activities to learn about rural people such as discussions of public health from the perspectives of person-centered care and community orientation. To promote learning, the participants reflect on their performance through 10–15-minute-long discussions with their teachers at the end of each day [23].

### 2.3. Participants

The participants were rural CBME stakeholders at Unnan City Hospital, which educates more than 40 trainees in family medicine every year [20]. Participants included physicians (the dean, the director of the hospital, and medical educators), nurses (the director of nursing, deputy directors, and head nurses), clerks, and citizens collaborating in the CBME.

### 2.4. Measurement

First, direct observation was performed by the first author, a family physician who has worked and been in charge of medical education at Unnan City Hospital for five years. The author observed the clinical situations at and medical education of undergraduate and postgraduate trainees in Unnan City Hospital from 1 April to 31 May 2020. Field notes were taken through direct observation and by conversing with the dean, medical doctors, medical teachers, nurses, medical students, a medical clerk, and rural citizens in the hospital and surrounding rural communities. Field notes focused on the interactions between healthcare professionals, citizens, and medical trainees with regard to the trainees’ behaviors, conversation style, and medical training at the hospital. One-on-one interviews, each about 20 minutes long, were conducted with each participant from April 1 to September 30 2020. The interviews were recorded and their contents transcribed verbatim. The authors analyzed the transcriptions based on a thematic analysis. The interview guide included four main questions: What did you think about when accepting medical trainees in this pandemic? What do you think about the inhibitions regarding accepting medical trainees in this pandemic? What do you think about the drivers of accepting medical trainees in this pandemic? What do you think about rural CBME for your hospital?

### 2.5. Data Analysis

Thematic analysis was used to identify the challenges posed by the COVID-19 pandemic and the possible solutions for sustainable rural CBME in community hospitals in Unnan [24]. The first and second authors carefully read the field and reflection notes and interview transcriptions. The first author then coded the content and developed codebooks based on repeated reading [24]. The second author also coded the materials and discussed the coding and codebooks with the first author. In this process, the authors inducted, merged, deleted, and refined concepts and themes by comparing research materials and coding [7]. Discussion of data and coding continued until mutual agreement was reached and no new concepts and themes emerged. For member checking, the analysis was provided to all participants, whose feedback was included in the final revision of the themes and concepts. Eventually, no new themes emerged during the member checking phase, indicating saturation. Finally, the themes and concepts were discussed and agreed upon by all authors.

### 2.6. Ethical Considerations

Before providing written consent, participants were informed that the data would be used for research purposes only. They were also informed about the research aims, data disclosure procedures, and steps taken to protect personal information. This study was approved by the Unnan City Hospital Clinical Ethics Committee (Approval code 20200018).

## 3. Results

Overall, 31 pages of field notes were taken through direct observation. In all, six physicians, eight nurses, three clerks, fourteen medical trainees, and three citizens were interviewed. Through the thematic analysis, three themes emerged: uncertainty surrounding the COVID-19 pandemic, an overwhelming fear within rural communities, and motivation and determination to continue providing CBME. Figure 1 depicts the conceptual framework.

### 3.1. Uncertainty of COVID-19

Most COVID-19 cases have been reported in urban populations, with few cases in rural areas. After a state of emergency was declared in Japan, undergraduate and postgraduate medical education institutions became hesitant about exposing their trainees to patients, even those with no COVID-19 symptoms. One of the participants stated:

“In this pandemic, there are many patients who are asymptomatic, and young patients are spreading their infections in urban areas. So, medical trainees can be transmitters of the infection even in rural areas. That is why at first, we did not welcome medical trainees.” (Physician 5).

An abundance of caution was observed in rural settings, where, ironically, COVID-19 was not a major concern. Medical professionals in the community hospital feared the ambiguity associated with these infections and the risk they pose. One of the participants stated:

“The present situation is complicated because of a lot of unknown things regarding COVID-19. There are various rumors about the transmission of COVID-19, which can cause anxiety among medical staff and citizens.” (Nurse 1).

CBME for medical trainees was not accommodated, even in rural settings, where the hospitals did not have many patients with COVID-19. COVID-19 can be transmitted via air droplets and asymptomatic patients can spread the virus to others, although no studies have conclusively confirmed the latter possibility. One of the participants stated:

“This infection has many unknowns. So, we tend to feel more fear than necessary. The additional activities arising from this fear may overburden us, such as in medical education.” (Clerk 1).

Furthermore, medical trainees feared potentially transmitting the infection to communities, even when they did not have any evidence of being infected. One of the participants stated:

“I know I am not infected with COVID-19, because there is no patient in my prefecture and I have never been exposed to suspected patients. But there is no certainty regarding this infection.” (Medical trainee 1).

The rapid spread of the infection changed the role of the rural hospital, which had to prepare quickly to accept patients with COVID-19. The speed at which this change had to take place left healthcare professionals exhausted. In addition, despite low risk, medical education activities were restricted by the Unnan City Hospital administration owing to concerns that trainees could be exposed to infected patients from outside the rural city. Rural healthcare professionals hesitated to accept outsiders, including medical trainees. One of the participants stated:

“Our hospital’s function may change through this pandemic. Although we do not know our hospital’s role clearly, we have to prepare for changes. In such a situation, education of trainees can be excluded from the mainstream hospital administration activities.” (Physician 3).

### 3.2. Overwhelming Social Fear and Pressure within and Outside Communities

The pandemic has perplexed medical educators and administrators of institutions that engage in medical education. With public accusations of COVID-19 transmission among the staff of medical institutions, administrators are apprehensive about involving medical students in patient management. The fear among stakeholders, such as deans of teaching hospitals, healthcare professionals, and the general public, has resulted in the inhibition of CBME. There is a perception that even a single COVID-19 case could be detrimental to both hospital conditions and their reputations and that all staff dealing with cases of COVID-19 might risk transmitting the infection to their families. This has led to unfavorable circumstances, such as Unnan City Hospital being unwilling to accept medical trainees and residents. Some of the participants stated:

“COVID-19 appeared suddenly, and our rural communities had several cases. We cannot focus on the future of COVID-19 clearly. If there is a transmission of this infection to our staff, we may have to close our hospital. So, we tend to hesitate to accept medical trainees from other institutions…My family can get infected if I transmit the virus from the hospital to my home. When I imagine such a situation, I am devastated because my family can be ostracized by society.” (Clerk 2).

“We are experiencing rejection from the rural communities. We do not have any symptoms, and there are no patients in our prefecture. However, just coming from urban areas can trigger people’s fear of COVID-19, which can inhibit acceptance of learning in communities.” (Medical trainee 7).

The COVID-19-induced social fear has been growing stronger as the number of patients increases. In rural areas, however, where the number of patients with COVID-19 is low, there is a possibility that residents could have neglected to take appropriate precautions. However, as social fear spread and other citizens’ perceptions began to be considered, they became hesitant about accepting medical trainees. One of the participants stated:

“Rural social norms and fear are persistent even when there are no patients with COVID-19. We can accept medical trainees, and they can motivate us to work in communities by being interested in our lives and traditions. However, when thinking about the continuity of our lives, accepting medical trainees can be a risk.” (Citizen 2).

In rural CBME, medical trainees participate in community activities to motivate rural citizens to live actively. During this pandemic, while there were some activities in rural communities and though rural citizens suggested that medical trainees could participate, the community hospital remained hesitant. Staff involved in CBME feared the possibility of medical trainees transmitting the infection to rural citizens, which could weaken the reputation of the hospital. One of the participants stated:

“Community rumors can affect our hospital’s reputation. On social media, hospitals with intra-hospital COVID-19 infection were criticized drastically, and our hospital could be next. Many stakeholders in our hospital may hesitate to accept medical trainees and provide CBME.” (Clerk 1).

The pandemic situation is constantly evolving because of new findings; this uncertainty causes fear in the stakeholders of rural CBME. This had led to various negative predictions for the future of education in rural institutions. This vicious cycle inhibits the provision of CBME in rural medical institutions.

### 3.3. Motivation and Determination to Continue Providing CBME

Unnan City Hospital is dependent on physicians who are educated there—it is important to ensure that medical students continue to be trained as family physicians. Through rational discussions about family medicine and the risk of COVID-19 infection, stakeholders acknowledged that inhibiting CBME may reduce the number of family physicians, which would further propel fear. Some of the participants stated:

“We have to prepare for the situations that may arise after this pandemic, and educating medical trainees can be important for the future of the hospital and the rural communities.” (Physician).

“There are some risks of infection in this era. However, the risk of losing future medical resources should be considered regarding not only physicians but also nurses.” (Nurse 4).

Some staff at the hospital were determined to continue with CBME despite the risks because it was deemed necessary for the sustainability of the community’s healthcare. One of the participants stated:

“Medical educators and trainees’ motivation can be respected for the hospital’s future. Our CBME can improve the hospital’s medical care and lead to an increase in the number of family physicians. This pandemic may prolong further, and the continuity of CBME should be discussed to ensure sustainable community care.” (Nurse 6).

Medical educators and trainees were determined to maintain their health and learning during this pandemic. This was encouraged through continuous discussion among stakeholders regarding the benefits of CBME for family medicine. Involving stakeholders in CBME provision was vital for the sustainability of CBME at Unnan City Hospital [22]. The participants stated:

“In rural CBME, we can experience a lot of clinical situations and learn the importance of family medicine and primary care. In the situation of university and general hospital education, clinical experience in rural hospitals is vital for medical trainees to increase clinical experiences.” (Medical trainee 5).

“As medical educators, we can teach medical trainees through collaboration with various stakeholders. This time, we hesitated to accept medical trainees initially, but thinking about the future of rural communities, we saw that CBME can be critical for sustaining medical resources.” (Physician 5).

This also resulted in standards being set for accepting medical trainees and the extent to which the trainees could interact with patients, based on the severity of the pandemic. The driver of this discussion was the realization of the importance of CBME for the hospital’s future. Medical educators at the rural hospital recognized the limitations of CBME in this pandemic but remained strongly motivated to educate trainees. Everyone involved developed a strong determination to continue CBME. One of the participants stated:

“For our hospital’s future, CBME should be continued. This education is sustaining our hospital’s function. For the provision of CBME during this pandemic, we have to share the vision of CBME in this hospital. The provision of CBME for medical trainees and our hospital’s future should be established through a discussion among stakeholders.” (Physician 1).

## 4. Discussion

This study shows the present conditions of rural CBME and the need for sustaining CBME in rural community hospitals during the COVID-19 pandemic. The lack of evidence regarding the control of COVID-19 [25] has induced various fears in the stakeholders of medical education. It was evident that CBME provision should be determined by both the needs of medical students and universities and the concerned hospital’s consideration of its future as well as the future of the community it is responsible for [26]. Furthermore, social norms in communities should be respected with regard to COVID-19. Local perceptions of medical education in pandemic conditions affect the acceptance and effectiveness of rural CBME. To prepare for future confusion, rural stakeholders should be respected and supported based on their determination to continue providing CBME.

The fears regarding accepting medical trainees from urban areas do not disappear even if stakeholders’ motivation and determination remain strong. COVID-19 has spread worldwide, and researchers are dedicatedly studying this virus, leading to a large volume of scientific publications. New information is constantly being disseminated by the mass media with updates on the standards of managing the infection [27]. As the participants stated, the situation of the pandemic is changing constantly and the rapidity with which the changes are occurring is overwhelming. They also deeply feel the risk of infection and becoming carriers of the virus. Based on the participants’ statements, accepting medical trainees from urban areas can trigger fear, inhibiting their work through increased infection control work. The trend of fearing medical trainees can be strong among non-medical professionals such as medical clerks and care workers because of a lack of medical knowledge [28]. Constant provision of information and mitigation of stress in non-medical workers should be performed frequently for the continuity of rural CBME. This can be done by sharing information about the pandemic and providing clarifications about the risk of infection they may face from medical trainees using statistics. 

The balance between rural CBME and the severity of the pandemic is essential for the continuity of rural CBME. There may be several waves of the COVID-19 pandemic, as well as fluctuations in the number of patients and the risk of infection [29]. As stated by the participants, medical educational institutions have to adjust their modes of teaching based on the pandemic situation. Even though rural social norms should be respected during the pandemic, the dialogue with various stakeholders is essential for the continuity of CBME [26]. Besides, this study’s analysis shows that rural CBME is critical for medical trainees to gain experience in primary care and family medicine. Interactions with rural citizens improve medical students’ motivation to learn rural medicine and become family physicians [30]. Establishing a standard for accepting medical trainees and the range of clinical training allowed in community hospitals should be discussed among stakeholders, including the medical trainees themselves [31]. This process can motivate medical trainees and medical universities that allocate students to institutes to consider tangible measures to provide rural CBME. Furthermore, providing education via video-based platforms during peak transmission should be considered, and to assuage people’s fears regarding interacting with medical trainees during CBME, providing adequate precautions and counseling regarding COVID-19 is essential.

Stakeholders’ motivation and determination can be sustained through continuous discussion of the importance of rural CBME for the sustainability of rural medicine. Through this pandemic, various stakeholders have experienced tremendous difficulties in imparting rural CBME [32]. Furthermore, deans and clerks in community hospitals have had to negotiate the criteria of acceptance of medical trainees with the hospitals and universities that allocate medical trainees to them. Referring to the previous article related to rural Japanese contexts of the COVID-19 pandemic, medical trainees and medical teachers struggled to learn clinical reasoning owing to the corruption of pretest probabilities of diseases because of the fear of COVID-19 [18]. They had to consider the possibility of being infected because of the continuous exposure to patients [33]. As stated by the participants, rural nurses also struggled with constructing the system of infection control for COVID-19 because they were not used to performing infection control while accepting medical trainees [34]. Each stakeholder in rural CBME has had to focus on different issues in this pandemic. For the effective continuity of rural CBME, there must be continuous dialogue and discussion where each stakeholder’s challenges and ideas are respected [9,14]. In fact, their mutual understanding and effective collaboration in the administration of rural CBME can drive the smooth adaptation of the education system in this pandemic [35]. Furthermore, continuing on-site learning through an experiential learning program during the pandemic can improve medical students’ motivation and reflection on clinical medicine [36]. Based on the previous articles, medical students who are motivated to learn in rural areas can work in rural medical institutions [37,38], but the quality of rural CBME is critical for medical students’ motivation [39]. During the dialogue, evidence of the importance of rural CBME for the sustainability of rural medicine and the increase in the number of rural physicians in rural settings should be discussed, as this can promote the motivation and determination to impart rural CBME.

Despite its strengths, this study has certain limitations. First, as this study was conducted at a single rural Japanese community hospital, the findings have limited transferability. However, the educational system has been described in depth, which can facilitate application to other settings. In addition, this pandemic can distort the perception of rural hospitals and CBME; therefore, our results can be applicable to the continuity and management of their CBME systems. The second limitation is confirmability. This study was conducted mainly by a medical educator in the community hospital, and the relationship between the interviewer and interviewees might have affected the contents of the interviews. To mitigate these limitations, the first researcher discussed the contents with the third author, who is a specialist in infection control and is not from the hospital. To improve the confirmability and transferability of this study’s findings, following studies can address the limitations of potential interviewer biases by bringing in external interviewers. Furthermore, following studies can investigate the effectiveness of rural CBME on medical students during this pandemic both qualitatively and quantitatively, which can show the whole picture of the effective adaptation of CBME during the pandemic.

## 5. Conclusions

Rural CBME was impinged during the pandemic by not only the fear of infection but also social fear and pressure within and outside communities. However, CBME in rural areas can continue to serve its true purpose if the necessary precautions are given careful consideration and continuous dialogue between stakeholders and medical educators based on rural social norms and culture is ensured. To ensure the sustainable provision of CBME in rural settings, constant assessment of the risks associated with the pandemic and the implications for CMBE is required. This is essential not only for medical educators and students but also stakeholders who administrate rural CBME.

## Figures and Tables

**Figure 1 healthcare-09-00223-f001:**
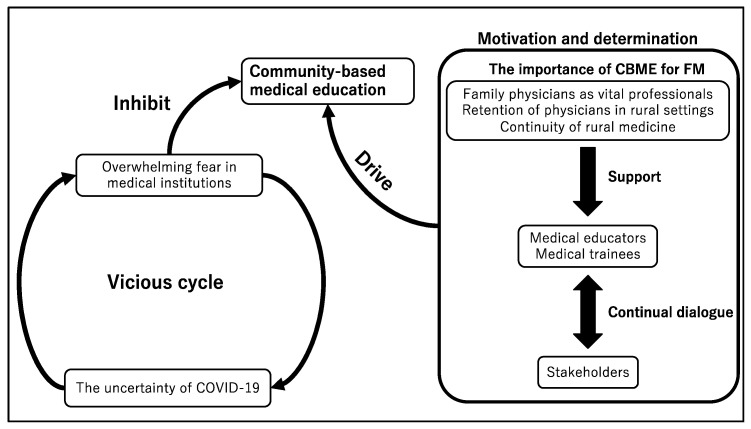
The vicious cycle of the fear of coronavirus disease 2019 (COVID-19) and the challenge to the continuity of CBME. FM—family medicine.

## Data Availability

All relevant datasets in this study are described in the manuscript.

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
