# Peer review of "The Uncertainty of COVID-19 Inducing Social Fear and Pressure on the Continuity of Rural, Community-Based Medical Education: A Thematic Analysis"

_healthcare, 2021, doi:10.3390/healthcare9020223_

Round 1
Reviewer 1 Report
I feel that the paper is of solid quality overall. The following are questions for the authors to consider. In part, they reflect some of my recent experience in COVID-19 related crisis interventions in healthcare:
(1) Some attention should be paid to the "home" factor of medical professionals. This includes the families if the doctors, nurses, etc. There might be concerns of bringing infection from the home into the hospital, and from the hospital back home. The home might include children, partners, and parents. I did not see that part represented.
(2) To what extent are the medical schools in Japan partners in the training (in the classroom) for CBME? Without knowing much about medical schools in Japan are there "tracks" in those schools to send students out to rural community hospitals, urban community hospitals, and academically affiliated "teaching hospitals" (like we have in the US). My point is that the medical schools would seem to be a significant stakeholder in this discussion.
(3) Regarding (2) above, I would recommend that the authors provide some empirical data to demonstrate how there might be a shortage of medical professionals in rural areas as well as the % of medical school graduates taking positions in urban versus rural areas. That would support the notion of a shortage and perhaps help develop workforce planning strategies in healthcare to incentivize people to go to rural areas (at least for a specific period of time).
Author Response
I feel that the paper is of solid quality overall. The following are questions for the authors to consider. In part, they reflect some of my recent experience in COVID-19 related crisis interventions in healthcare:
- Some attention should be paid to the "home" factor of medical professionals. This includes the families if the doctors, nurses, etc. There might be concerns of bringing infection from the home into the hospital, and from the hospital back home. The home might include children, partners, and parents. I did not see that part represented.
Response:
Thank you for the positive evaluation of our paper. Regarding the Results, we have addressed your concern by adding a quote about the home situation of hospital staff (lines 199–203).
- To what extent are themedical schools in Japan partners in the training (in the classroom) for CBME? Without knowing much about medical schools in Japan are there "tracks" in those schools to send students out to rural community hospitals, urban community hospitals, and academically affiliated "teaching hospitals" (like we have in the US). My point is that the medical schools would seem to be a significant stakeholder in this discussion.
Response:
Medical schools are a significant stakeholder in this discussion, and we have revised the Introduction to clarify this (lines 58–60).
- Regarding (2) above, I would recommend that the authors provide some empirical data to demonstrate how there might be a shortage of medical professionals in rural areas as well as the % of medical school graduates taking positions in urban versus rural areas. That would support the notion of a shortage and perhaps help develop workforce planning strategies in healthcare to incentivize people to go to rural areas (at least for a specific period of time).
Response:
We have added descriptions regarding the rate of physicians working in rural areas and students’ motivation to work in rural areas (lines 33–35).
Reviewer 2 Report
1. Grammar throughout the paper can be improved
2. Introduction section could be improved, especially regarding review of previous and related literature and how that influences and compares the current study.
3. Methods and Results have been well presented
4. Discussion section can be improved. Specifically, while the author clearly presents the impact of the pandemic on CBME, they can further the discussion by providing meaningful suggestions for handling the situation. e.g. providing education via video based platforms during peak transmission, addressing fear in the students by providing adequate precautions and counseling ,etc.
Author Response
- Grammar throughout the paper can be improved
Response:
We have made the necessary revisions throughout the manuscript.
- Introduction section could be improved, especially regarding review of previous and related literature and how that influences and compares the current study.
Response:
We have revised the section on related literature and the COVID-19 pandemic’s influence on CBME.
- Methods and Results have been well presented
Response:
Thank you for the positive evaluation of the Methods and Results sections.
- Discussion section can be improved. Specifically, while the author clearly presents the impact of the pandemic on CBME, they can further the discussion by providing meaningful suggestions for handling the situation. e.g. providing education via video based platforms during peak transmission, addressing fear in the students by providing adequate precautions and counseling ,etc.
Response:
We have made the indicated revisions to the Discussion (lines 314–317).
Reviewer 3 Report
The authors have highlighted the current conditions of rural CBME as impacted by the COVID-19 pandemic and the need for sustaining the medical training in rural hospitals. This is an important study given the current scenario and uncertainty of the future.
A few comments:
1) A figure showing the number of CBME students each year (maybe in the past 10 years) and the decline this year due to the pandemic, would make a stronger case.
2) The potential for having the students in a bubble in the rural areas where they will be serving should be discussed. There is no information if that was an option. This could be one way that the students and the rural areas could mutually benefit.
3) Throughout the manuscript, the authors mention several times that there were minimal to no cases in the rural areas, but in line 196 they say “COVID-19 appeared suddenly, and our rural communities had several cases. We cannot 196 focus on the future of COVID-19 clearly. If there is a transmission of this infection to our staff, we 197 may have to close our hospital. So, we tend to hesitate to accept medical trainees from other 198 institutions.” (Clerk 2). The author should check for inconsistencies and clarify.
4) It may also be a good idea to have a table or figure for the recommendation/solution to the scenario discussed.
Author Response
The authors have highlighted the current conditions of rural CBME as impacted by the COVID-19 pandemic and the need for sustaining the medical training in rural hospitals. This is an important study given the current scenario and uncertainty of the future.
A few comments:
1) A figure showing the number of CBME students each year (maybe in the past 10 years) and the decline this year due to the pandemic, would make a stronger case.
Response:
Medical schools and students are a significant stakeholder in this discussion, and we have revised the Introduction to clarify this (lines 58–60).
2) The potential for having the students in a bubble in the rural areas where they will be serving should be discussed. There is no information if that was an option. This could be one way that the students and the rural areas could mutually benefit.
Response:
We have added descriptions regarding the rate of physicians working in rural areas and students’ motivation to work in rural areas (lines 33–35).
3) Throughout the manuscript, the authors mention several times that there were minimal to no cases in the rural areas, but in line 196 they say “COVID-19 appeared suddenly, and our rural communities had several cases. We cannot 196 focus on the future of COVID-19 clearly. If there is a transmission of this infection to our staff, we 197 may have to close our hospital. So, we tend to hesitate to accept medical trainees from other 198 institutions.” (Clerk 2). The author should check for inconsistencies and clarify.
Response:
We have made the indicated revisions to the result (lines 195–205).
4) It may also be a good idea to have a table or figure for the recommendation/solution to the scenario discussed.
Response:
We have made the indicated revisions to the Discussion (lines 331–340).
This manuscript is a resubmission of an earlier submission. The following is a list of the peer review reports and author responses from that submission.
Round 1
Reviewer 1 Report
Dear Authors,
thanks for proposing this contribution regarding SARS-CoV-2 pandemic.
- Please contextualize better the current pandemic scenario;
- Please provide number and documentation about the approval by the ethical committee and report it in the text;
- healthcare student have a two-face nature, because their internship that can be seen as an opportunity for high-risk infection. Please consider and investigate or deal this aspect deeply;
- which criteria for the selection of psychometric tools for the one-to-one interviews? Please provide the questionnaire used and explain if is a validated tool or a tailored one containing different validated tools;
- deal with gender;
- conclusion must be improved, too poor. What are the possible repercussions? What suggestions to give to the health policy maker and stakeholders?
- you must consider we are facing a CoViD-19 second flow and you're paper is not up to date, but you should deal this in the introduction with the latest data regarding the second flow and consider it into discussion and conclusion sections;
- you did not investigate the fear of being stigmatized or discriminated against if they test positive or a family member / close cohabitant tests positive for SARS-CoV-2. This is a limitation that you may consider and deal with;
- what about the risk perception?
Please update these gaps referring to the following references:
- Sauer, K.S.; Jungmann, S.M.; Witthöft, M. Emotional and Behavioral Consequences of the COVID-19 Pandemic: The Role of Health Anxiety, Intolerance of Uncertainty, and Distress (In)Tolerance. Int. J. Environ. Res. Public Health 2020, 17, 7241
- Baldassarre, A.; Giorgi, G.; Alessio, F.; Lulli, L.G.; Arcangeli, G.; Mucci, N. Stigma and Discrimination (SAD) at the Time of the SARS-CoV-2 Pandemic. Int. J. Environ. Res. Public Health 2020, 17, 6341
- Ding Y, Du X, Li Q, Zhang M, Zhang Q, Tan X, et al. (2020) Risk perception of coronavirus disease 2019 (COVID-19) and its related factors among college students in China during quarantine. PLoS ONE 15(8): e0237626
- Sarah Dryhurst, Claudia R. Schneider, John Kerr, Alexandra L. J. Freeman, Gabriel Recchia, Anne Marthe van der Bles, David Spiegelhalter & Sander van der Linden (2020) Risk perceptions of COVID-19 around the world, Journal of Risk Research, DOI: 10.1080/13669877.2020.1758193
- Wise, T., et al. (2020) Changes in risk perception and self-reported protective behaviour during the first week of the COVID-19 pandemic in the United States. Royal Society Open Science. doi.org/10.1098/rsos.200742
Latest news about vaccination call an issue presented by WHO in 2019, vaccine hesitancy; there's a good assist to deal with.
Reviewer 2 Report
I enjoyed reading this article. The issue with allocating students in health programs to appropriate rotations during Covid-19 is indeed a problem worldwide. However, while the paper is interesting, there are several limitations.
1) more detailed information on the process used to determine the themes should be included. Usually, there are at least three independent authors that analyze the data collected.
2) the article is based in one community in rural Japan so it is not too relevant to other readers.
3) the authors should provide data (numbers) of students how were allocated before covid-19. It is not clear how big the issue is.
4) while covid-19 has impacted the training of future health professionals, sooner or later the overall situation will improve.
5) while at the beginning of the pandemic, all student related learning activities inside hospitals worldwide were halted, with time, solutions were found.
6) the article describe mostly problems but does not provide well-supported solutions